# Critical Success Factors for Successful Implementation of Healthcare 4.0: A Literature Review and Future Research Agenda

**DOI:** 10.3390/ijerph20054669

**Published:** 2023-03-06

**Authors:** Michael Sony, Jiju Antony, Guilherme L. Tortorella

**Affiliations:** 1WITS Business School, University of Witwatersrand, Johannesburg 2158, South Africa; 2Oxford Brookes Business School, Oxford Brookes University, Oxford OX3 0BP, UK; 3Department of Industrial and Systems Engineering, Khalifa University, Abu Dhabi P.O. Box 127788, United Arab Emirates; 4Mechanical Engineering Department, The University of Melbourne, Melbourne, VIC 3010, Australia; 5IAE Business School, Universidad Austral, Buenos Aires B1630FHB, Argentina; 6Production Engineering Department, Universidade Federal de Santa Catarina, Florianopolis 88040-900, SC, Brazil

**Keywords:** healthcare 4.0, smart hospitals, digital transformation, health 4.0

## Abstract

The digitization of healthcare services is a major shift in the manner in which healthcare services are offered and managed in the modern era. The COVID-19 pandemic has speeded up the use of digital technologies in the healthcare sector. Healthcare 4.0 (H4.0) is much more than the adoption of digital tools, however; going beyond that, it is the digital transformation of healthcare. The successful implementation of H 4.0 presents a challenge as social and technical factors must be considered. This study, through a systematic literature review, expounds ten critical success factors for the successful implementation of H 4.0. Bibliometric analysis of existing articles is also carried out to understand the development of knowledge in this domain. H 4.0 is rapidly gaining prominence, and a comprehensive review of critical success factors in this area has yet to be conducted. Conducting such a review makes a valuable contribution to the body of knowledge in healthcare operations management. Furthermore, this study will also help healthcare practitioners and policymakers to develop strategies to manage the ten critical success factors while implementing H 4.0.

## 1. Introduction

The healthcare industry is undergoing drastic changes due to an acceleration in costs to the extent of $600 billion worldwide by 2027, making healthcare less affordable and threatening the sustainability of industry margins [1]. However, there are opportunities to the extent of $1 trillion to create value, thus improving healthcare by raising clinical productivity, transforming the delivery of care, simplifying administrative procedures, and applying Industry 4.0 technologies [1,2]. Accordingly, there is an urgent need to digitize healthcare services so that the sustainable development goals of healthcare can be met [3]. The fourth industrial revolution has drastically impacted healthcare services, resulting in marked changes in the manner in which healthcare services are delivered using technologies. To cite some examples of how the healthcare industry has seen a significant transformation through the adoption of technologies, 3D printing has emerged as a promising tool for offering new solutions to patients. Ref. [4] have reported that 3D printing technologies have enabled the creation of prosthetics, surgical devices, and customized replicas of bones, organs, and other anatomical structures. In a similar vein, virtual reality technologies have been utilized in diagnoses and treatment, medical training and education, teleconferencing, dentistry, psychiatry, surgery, and cognitive rehabilitation [5]. Robotics has also played a crucial role in the healthcare industry, particularly in surgery and minimally invasive surgery [6,7]. Beyond that, robots are used in fields such as assisting disabled individuals [8] and counselling, where the human touch element is essential [9]. Additionally, technology has been integrated into anatomical models, drug formulations, regenerative medicine, and engineered tissue models to improve treatment success rates [10]. These technological advancements have the potential to significantly transform the healthcare industry and improve patient outcomes. Thus, the healthcare system is transforming into a smart, integrated system with improved personalized care, high service quality, better customer experience, better outcomes, and lower costs. As a result, there has been a paradigmatic shift in healthcare systems: (1) toward health management instead of the treatment of disease, (2) where clinical outcomes and quality are being given more importance, (3) moving toward outpatient services (i.e., the retailization of health services), (4) with informed and knowledgeable patients, and (5) emphasizing accountability and a values focus [11]. Furthermore, the COVID-19 pandemic has also accelerated the adoption of digital technologies in hospitals to meet the needs of universal healthcare [12].

One of the definitions of healthcare 4.0 (H 4.0) is a continuous but disruptive process of changing the entire healthcare value chain, including the production of drugs and medical equipment, hospital care, nonhospital care, healthcare logistics, healthy living environments, financial systems, and social systems. A vast number of cyber and physical systems are closely combined through the Internet of Things, intelligent sensing, big data analytics, AI, cloud computing, automatic control, autonomous execution, and robotics to create digitized healthcare products, services, and enterprises [13]. The implementation of H 4.0 may result in the achievement of the 8-Ps in healthcare: (1) preventive, (2) patient-centered, (3) personalized, (4) precision, (5) participatory, (6) predictive, (7) pre-emptive, and (8) pervasive healthcare [14]. Despite its immense potential, the literature on H 4.0 has been scattered, lacking academic alignment and practical directions [15].

H 4.0 is not only a technical phenomenon but also a complex mixture of social (human) [16,17] and technical systems [15,18] intertwined in a goal-directed manner to meet the objectives of healthcare systems [19]. The term healthcare 4.0 has appeared on blogs and webpages for years; however, the first Scopus indexed research paper using the term “healthcare 4.0” surfaced in 2018 [14,20]. While the term healthcare 4.0 is now commonly used in academic and industry literature, its implementation has been limited [13,15,18]. Some of the reasons for limited implementation have been low acceptance of H 4.0 by various stakeholders [21], high cost, data fragmentation, privacy and security concerns, lack of a standardized framework [17,20,22], and a high failure rate of digital transformation initiatives in healthcare [23,24].

Thus, we need to understand what key areas, activities, or factors must be effectively executed for the successful implementation of H 4.0. The critical success factors (CSFs) are typically the most important things that must go well for H 4.0 to be successful. They are specific, measurable, and unique to the context under consideration, and identifying and focusing on CSFs can help a healthcare organization to prioritize its efforts and resources, make better decisions, and stay on track with implementing H 4.0. Thus, there is a need for a study that critically investigates the CSFs for the successful implementation of H 4.0. accordingly, the research questions this study investigates are: **RQ1:** What are the critical success factors for the successful implementation of H 4.0?**RQ2:** How should research on H 4.0 proceed given our findings?

To answer those questions, this study uses a systematic literature review to critically analyze the previous literature, to identify the critical success factors for successfully implementing H 4.0. The remaining part of the article first describes H 4.0, followed by the methodology, descriptive analysis, and thematic analysis for unearthing the 10 critical success factors, the scope for future research, and the conclusion.

## 2. Literature Review

Healthcare systems have undergone transitions over time [21]. Before the 20th century, healthcare was largely a cottage industry, with patients receiving care at home from family members, friends, or local healers. Healthcare 1.0 is devoted to patient–clinician encounters [18]. Here, the patient visits the clinic and meets the physician and other healthcare practitioners, seeking solutions to their health problem. The physician and his team through consultation, diagnosis, and testing provide treatment and subsequent follow-up. Healthcare is mostly reactive, with doctors providing treatment after a patient becomes sick. Healthcare 1.0 was prevalent for a long period [25]. The issues in this phase were a lack of focus on preventative care and public health measures, limited medical knowledge and technology, and one-on-one healthcare delivery, making it difficult to provide care to a large population. In the early 20th century, hospitals became more institutionalized and began using modern medical technologies such as X-rays, antibiotics, and surgical procedures. Healthcare became more centralized and professionalized and hence Healthcare 2.0 was an era of major development in health, life science and biotechnology, medical equipment, and devices [19]. This was an era where imaging testing equipment, monitoring devices, and surgical and life support systems were increasingly used in healthcare systems [18]. Hospitals became more prominent, and health insurance emerged. However, access to healthcare was still limited based on socioeconomic situations, and the quality of care varied widely. Furthermore, there was a lack of standardization in medical practices and limited understanding of public health and disease prevention. With the advent of computers and digital technology, healthcare became more data-driven and information-based and hence Healthcare 3.0 was the era of implementation of information systems, electronic health, and medical records to manage patient records across various units and departments of healthcare systems and introduce telemedicine. Many manual processes within the healthcare systems were digitized [26]. However, there were still issues with data privacy and security, and not all patients had access to these technologies, creating a healthcare divide based on socioeconomic situations. In addition, there was a need for more standardization and interoperability in health IT systems. H 4.0 is an era where healthcare delivery is enabled by (1) medical cyber–physical systems (MCPS), (2) RFID, (3) IoT, (4) medical robots, (5) wearable and ambient sensors, and (6) intelligent sensors, which are integrated with cloud computing, artificial intelligence, big data analysis, and decision support techniques, to achieve smart and interconnected healthcare delivery [18,25]. Here, all healthcare organizations such as primary, secondary, and tertiary healthcare facilities, equipment and device suppliers, patients, and communities are integrated into a digital ecosystem. In this era, patient care is forecast to shift in terms of the use of AI techniques [27] and big data analytics [28] for proactive treatment, disease prediction and prevention, personalized medicine, and enhanced patient-centered care [25,29]. H 4.0 will thus be an era of a pervasive, smart, and interconnected healthcare community, as depicted in Figure 1.

The two fundamental elements of H 4.0 are (1) smartness and (2) interconnectedness.

### 2.1. Smartness

Smartness is the use of Industry 4.0 technologies such as artificial intelligence and big data analytics to improve treatment, diagnosis, coordination, and communication between healthcare service providers, patients, and other stakeholders, to achieve patient-centered and individualized smart healthcare management [15]. This is achieved through the use of I 4.0 technologies as follows: (1) stratification and classification for patients for treatment, (2) prediction analysis for prediction of disease and development based on the previous phase, (3) preventive and proactive care using outputs from the previous phase, (4) monitoring, intervention, and optimal treatment to improve patient outcomes using I 4.0 technologies based on the previous phase, and (5) a closed loop [25,30]. All the elements and connected in closed loops so that dynamic improvement of care takes place as depicted in Figure 2.

### 2.2. Interconnectedness

This element depicts interconnection or integration across all aspects of the healthcare system. The basic idea is to form an effective information network by integrating: (1) interactions between caregivers, patients, and other team members responsible for patient care using I 4.0 technologies for integration, (2) digital technologies for managing all communications within the professional care team, (3) equipment and devices using sensors, IoT, and different kinds of integration devices, (4) organizations and community and other stakeholders, (5) insurance, billing, and costs as part of healthcare systems, and (6) care transitions across time and space, i.e., over a patient’s lifetime or between inpatient spaces, outpatient spaces, the home, and long-term care [25,31,32]. This is depicted in Figure 3.

Thus, smartness and information integration are desired across every stage of the patient journey, in every aspect of the healthcare system. A patient may stop at several healthcare facilities along the way, such as a primary care clinic, specialists, a hospital, an emergency room, a rehabilitation center, and long-term care, but the goal of H 4.0 is to create technologies that facilitate smart communication and cooperation amongst the various parties engaged in the patient journey.

## 3. Research Method

A systematic literature review (SLR) was conducted in this study to bring transparency and replicability to the research, to build a firm foundation for future research on H 4.0 [33]. The existing studies were critically analyzed to find the critical success factors and thus unearth areas for future research [34]. A successful SLR requires research questions and a subsequent SLR process [35]. The study used the process put forward by Ref. [36] to systematically carry out the literature review. The steps are detailed in Figure 4.

### 3.1. Data Sources

To design a search string for H 4.0, a previous literature review was used as a reference [15,17,37]. A three-part search string was designed. Part 1 AND part 2 AND Part 3 were used in all combinations as a search string. To source only good quality research papers, we used Web of Science, Scopus, Science Direct, Google Scholar, and PubMed as the databases. We excluded conference proceedings and articles that were not in English. The search string used is given in Appendix A.

### 3.2. Screening

In this phase, screening of the articles was conducted to eliminate systematic error or bias. This was carried out through a protocol suggested by Popay et al. [38]. This protocol is set out in Figure 5. The first step was an extensive search to find titles, abstracts, and keywords that met the screening criteria.

To maintain the quality of articles, we used Cabells’ list. If the article was from a predatory journal, it was eliminated [39]. After that, the titles of articles were examined, which helped to remove duplicates. The abstract was also read to determine whether it was suitable research. Furthermore, the reference list was combed to improve the search criteria. A breakdown of the stage-wise number of articles is shown in Figure 4.

### 3.3. Data Analysis

The main goal of this research was to determine the critical success factors of H 4.0. The final articles were read to identify directions, patterns, similarities, and differences [40,41,42,43]. Eighty-nine articles were included in our review, considering the research questions. The lead researcher along with three assistants engaged in the coding process. Beforehand, we conducted pilot coding, wherein we jointly designed instructions and a coding scheme. Eight articles were extracted randomly from the sample. The lead researcher along with a research assistant coded the articles for the themes of critical success factors for implementing H 4.0. These codes were compared, then a coding scheme and instructions were further developed. In the next phase, the remaining articles were coded, and themes were compared. The simple percentage agreement was calculated as 83%. Wherever there was disagreement, it was settled through discussion so that all viewpoints were captured.

## 4. Results

The analysis was conducted in two dimensions: (1) descriptive and (2) thematic analysis. The descriptive analyses were conducted to understand the evolution of the body of knowledge. The thematic analyses were conducted to unearth the critical success factors (CSFs) of H 4.0.

### 4.1. Descriptive Analysis

#### 4.1.1. Timeline Distribution

A time trend analysis of the articles was performed to reveal their distribution over time [44,45]. Figure 6 depicts the timeline distribution of the articles.

Articles on H 4.0 appeared in the last five years. There has been an increasing trend, suggesting that there has been increasing interest in the field of H 4.0.

#### 4.1.2. Number of Authors

Figure 7 indicates the number of authors for the articles published, according to our time trend analysis. Sixty-seven percent of the articles were authored by four or more authors, suggesting that H 4.0 is an emerging subject, on which authors are collaborating to develop the body of knowledge over time. This is understandable, as being an interdisciplinary subject, it requires the collaboration of authors from various disciplines to develop the body of knowledge.

#### 4.1.3. Productive and Influential Authors

The top ten most productive authors in this field are listed in Table 1. The productive authors were classified based on the number of publications.

Flavio S. Fogliatto from Universidade Federal do Rio Grande do Sul, Porto Alegre, Brazil was the most productive author with 12 publications. He was followed by 11 publications from Neeraj Kumar of the University of Petroleum and Energy Studies, India, Sudeep Tanwar from Nirma University, India, and Guilherme Luz Tortorella from the University of Melbourne, Australia.

#### 4.1.4. Influential Authors

To determine the most influential authors, we calculated the CPP (citations per publication), as are shown in Table 2.

In terms of the most influential authors, Parekh K. and Mistry I., though they had only one publication each, had better citations than the other authors. However, in terms of total citations, Tanwar S. and Kumar N. had higher numbers of citations, with 11 documents each.

#### 4.1.5. Top Source Journal

The top ten journals to have published articles on H 4.0 are depicted in Table 3.

The IEEE Journal of Biomedical and Health Informatics and IEEE Transaction on Industrial Informatics top the list.

#### 4.1.6. Countries

The top ten most productive countries in H 4.0 research are listed in Table 4. India tops the list with 34 documents, followed by Brazil and Australia. We infer that since India and Brazil are developing countries, researchers there have realized the importance of H 4.0 for their country’s development, leading to research articles from these countries.

#### 4.1.7. Keyword Analysis

Vosviewer was used to identify the top key words used by the authors, as depicted in Figure 8. H 4.0 is surrounded by technologies such as the Internet of Things, medical computing, electronic health records, artificial intelligence, Industry 4.0, digital storage, healthcare, and so on. Furthermore, it is applied in both diagnosis and patient treatment to improve healthcare service delivery and performance.

#### 4.1.8. Type of Study

The articles were classified into empirical and conceptual studies, as shown in Figure 9. The conceptual articles were articles that introduced concepts and ideas without testing them. The empirical studies were those that involved the use of empirical data.

The timeline distribution of articles suggests that though research started with conceptual articles, empirical articles have dominated the research domain. The empirical studies have revolved around the implementation of a particular technology, e.g., an IoT-fog-based H 4.0 system using blockchain technology.

### 4.2. Thematic Analysis of Articles

The thematic analysis of articles highlighted 10 critical success factors for the successful implementation of H 4.0.

#### 4.2.1. Digital Integration and Interconnectedness of the Healthcare Ecosystem

H 4.0 works in an integrated and interconnected ecosystem [46] where platforms used by the government, private hospitals, other health agencies, insurance firms, pharmacies, and other stakeholder are digitally connected. Data sharing among these entities should be encouraged but consider the privacy rights of the patients [47]. Healthcare data must be protected by high security and privacy [48] and hence security standards such as HIPAA, COBIT, and DISHA have been developed. There is a need for all stakeholders to use such standards in a uniform manner, or else there will be an issue concerning the interoperability of the systems [49]. Thus, for the successful implementation of H 4.0, the platforms of all stakeholders should be able to communicate with one another [50]. Another factor to consider is the technology infrastructure, which will enable connectivity among the stakeholders [51]. This will be a country-specific issue as some countries have better technology infrastructures than others. In addition, the level of digital connectivity will vary considerably between developed and developing countries. Regardless, however, for the success of H 4.0, a highly integrated and interconnected healthcare ecosystem is required.

#### 4.2.2. Human-Centric Automation of Healthcare Providers

Healthcare providers are labor-intensive and high in touch dimensions [17]. However, healthcare providers are using technology to upgrade their operations and automate workflows. This has resulted in the automation of various processes of healthcare providers, thus improving the productivity, efficiency, and efficacy of patient care [52]. Various functions in a hospital such as finance, medical, legal, administration, and compliance departments are automated to various degrees. Automation is also extended to inpatient and outpatient management, laboratories, pharmacies, radiology, blood banks, biometric systems, alert systems, feedback, HVAC, lighting, and so on [52,53]. There are various interfaces in a healthcare ecosystem. To name just one, patient–system interfaces could be automated by incorporating voice, gesture, and easy-to-operate touchscreen systems [54]. In addition, intelligent sensing and monitoring systems for patients will help in offering personalized care, made possible thanks to the high degree of automation of various patient activities [55,56]. Further monitoring of multiple advanced patient parameters will help in understanding the real-time status of patients and will help in the management of disease [57,58]. The success of H 4.0 implementation will depend on the degree of human-centric automation of healthcare providers [14].

#### 4.2.3. Improve Patient-Centricity and the Patient Experience

Patient centricity is defined as a dynamic process where the patient controls the flow of information to and from them through a variety of channels, enabling them to make decisions in line with their preferences, values, and beliefs [59]. This fundamentally transformative concept affects how healthcare decisions are made and who has the authority to make them [52,59]. Patients’ voices were suppressed for too long, but now patient-centricity gives not only a voice to patients but also emphasizes consideration of their values, thoughts, preferences, strengths, and shortcomings while making a healthcare decision [17,60]. In such ways, patient-centricity improves the patient experience. Modern technologies have also been used in hospitals to improve the patient experience [61]. Before the treatment phase, hospitals are using technologies such as wearable monitoring devices to render real-time personalized care [62]. During treatment, the digital identity of the patient is used, and medical records are generated and added to their data file, which can be assessed from anywhere. In addition, chatbots [63,64] and humanoids [65,66] are giving real-time help to patients throughout their treatment, in addition to the personalized care of physicians and support staff. They further help in patient education. After the treatment, wearable sensors offer real-time data about the patient [67] and help in extending care to patients’ homes and offices. Abnormal parameters are flagged, and the patient and their physician collaborate on the patient’s management [68]. In addition, technology will also help with setting reminders for follow-up and so on. The success of H 4.0 will be based on how digital technologies are used to improve patient-centricity and the patient experience.

#### 4.2.4. Use Big Data and Analytics

The medical IoT is being increasingly used in H 4.0, with data from devices stored considering the privacy and security of patients [69,70] Another factor to consider is the use of a medical cyber–physical system (MCPS), which combines embedded software control devices, complex physiological dynamics of patients, and networking capabilities in the modern medical field [22]. This results in medical cyber–physical data which are generated, digitally stored electronically, and accessed remotely by medical staff or patients [71]. The data can be used through analytics to improve every aspect of hospital operations and patient care [72]. Data analytics used in patient care are primarily in these dimensions: (1) prediction of disease progression, (2) early disease detection, (3) personalization of healthcare, (4) personalized management of disease, (5) drug discovery, and (6) managing patient data. They can also be used in healthcare operations for (1) automation of various administrative processes, (2) managing patient flows, (3) insurance, (4) accurate costing, (5) capacity management, and (6) scheduling [18,50,53,54]. The hospital typically uses four types of data analytics: descriptive, prescriptive, predictive, and discovery analytics, in both patient care and managing hospital operations. Some of the healthcare data analytics technologies are: (1) AI tools, (2) cloud computing platforms, (3) blockchain networks, (4) health information exchanges, and (5) machine learning models [73,74,75,76,77,78,79]. Thus, the successful implementation of H 4.0 warrants the use of big data analytics in both patient care and hospital operations management.

#### 4.2.5. Managing Digital Healthcare Supply Chains

Digital healthcare supply chains refer to the use of digital technologies and data analytics to manage and optimize the flow of healthcare products and services [8,22]. These technologies can help to improve the efficiency, transparency, and reliability of supply chain processes, while also reducing costs and enhancing patient outcomes. While implementing H 4.0, healthcare organizations need to invest in the necessary technology infrastructure, data analytics capabilities, and talent, to manage the supply chains. Another facet to consider is digitally managing the intersection of healthcare and supply chain principles to create service designs in healthcare that are affordable, accessible, meet the healthcare needs of patients, and are patient-centric [80,81]. One emerging concept is retail medical clinics (RMCs). RMCs are located inside retail shops and provide an array of healthcare services at a very low cost compared to hospitals, doctors, or emergency rooms but at the same quality. This concept improves patients’ access and awareness and healthcare’s affordability. RMC brings the following advantages: (1) provides bundles of healthcare services in one set, (2) resolves healthcare supply and demand issues, (3) frees hospitals up for major procedures while RMC takes care of basic needs, and (4) places the retail, efficiency, and cost-effectiveness of services under one umbrella [82]. Furthermore, the integration of RMCs, hospitals, and other healthcare stakeholders will help to create a healthcare ecosystem that is responsive, effective, affordable, and offers patient-centric healthcare. Thus, for the success of H 4.0, there should be a proliferation of RMCs in both urban and rural areas under the umbrella of major retail giants offering a bundle of services and digitally integrated ecosystems.

#### 4.2.6. Strategies for Promoting H 4.0

Healthcare innovation is fundamentally driven by rising healthcare costs, technological breakthroughs in the field of science, medicine, and rapid digitization of healthcare [50,55]. For this innovation to be acceptable at the grassroots level, the stakeholders should formulate strategies to implement H 4.0 [15,18]. These strategies should target building a technical infrastructure to promote H 4.0. In addition, strategies should also consider means to improve the highly skilled manpower in these smart healthcare systems [53]. Plus, further strategies should be formulated to support the promotion of integration among all stakeholders in healthcare. In these ways, successful implementation of H 4.0 can be made possible with the formulation of strategies for promoting H 4.0.

#### 4.2.7. Promote a Culture for H 4.0

The organizational culture in a healthcare setting reflects the shared ways of thinking, behaving, and feeling [36]. There are three levels of organizational culture in healthcare. The first is the visible manifestation. This comprises visible aspects of culture as follows: (a) distribution of services and roles in organizations, (b) the physical facilities’ layouts, (c) the established pathways of care, (d) demarcation between staff groups in activities performed, (e) staffing practices, reporting arrangements, and dress codes, (f) reward systems, and (g) the local rituals and ceremonies that support approved practices [83]. The second level constitutes shared ways of thinking, which include the values and beliefs used to justify and sustain the visible manifestations. The third level then constitutes deeper shared assumptions, which are unconscious and unexamined underpinnings of day-to-day practice [84]. Healthcare organizations are varied in terms of specialty, with groups based on occupation and hierarchies in terms of professions and service lines. Some beliefs may be shared and common and others may be dominant in certain groups. The implementation of H 4.0 involves not only the implementation of technical aspects of ICT integration but also dealing with social (human) elements in the implementation of concepts such as patient-centricity and improvements to the human experience [83]. For H 4.0 to be successful, the values and belief about improving patient-centricity and the patient experience using digital technologies should be shared among all stakeholders in healthcare. There is a need to change the thinking from traditional service-provider-oriented thinking to patient-centric thinking. This hinges on the promotion of a H 4.0 culture.

#### 4.2.8. Healthcare Leadership

Health leadership is defined as “the ability to identify priorities, provide strategic direction to multiple actors within the health system, and create commitment across the health sector to address those priorities for improved health services” [85]. H 4.0 implementation builds on two components: (a) smartness and (b) integration within healthcare systems. This requires new ways of carrying out healthcare activities as it enables real-time customization of care to patients and professionals [54]. The complexity of healthcare systems is in their public and private health providers, primary healthcare systems, acute, chronic, and aged care provisions, retail clinics, and so on [86]. Thus, there is a need for a leader who will strategically provide insights to these multiple stakeholders and motivate them to undertake the path of H 4.0 implementation, providing direction for all actors within the healthcare system.

#### 4.2.9. Healthcare Employees’ Skills

Healthcare employees are key to the successful implementation of H 4.0. The implementation of H 4.0 changes the traditional ways of working with healthcare service providers [50]. It involves delivering care anytime and anywhere through the aid of digitization [69]. The use of technology in healthcare will further extend care beyond the walls of healthcare service providers, and hence, healthcare employees, in addition to traditional skills of healthcare, also need digital skills [20,70]. Another point to consider is how core technical healthcare skills will transform with the use of technology. To provide an example, in addition to diagnostic tests, the physician may also rely on big-data-based analytics for confirming and contrasting the diagnostic findings. Such developments warrant the acquisition of new skillsets for healthcare workers from physicians to administrative staff. Furthermore, on-demand healthcare, teleconcilium, telemonitoring, teleconsultation, remote access to equipment, tele appointments, and so on also require new skills of healthcare workers [50,53,87]. Thus, for the success of H 4.0, there is a need for constant deskilling, reskilling, and upskilling for all categories of healthcare workers. However, there is a global shortage of healthcare workers, and some such as specialists and nurses are in huge shortage, leaving them with little time for training in new technologies.

#### 4.2.10. Adoption of New Business Models

Platform-based business models bring ecosystem participants in a digital network together to co-create goods and services [19,87]. These models are making inroads into healthcare providers. Traditionally, mergers and acquisitions were used by healthcare service providers for increasing market share or mitigating threats [88]. The advances in technology in healthcare have resulted in using platform-based business models to achieve newer revenue avenues. H 4.0 provides numerous opportunities, and the need for building localized ecosystems may be harnessed by healthcare service providers. However, there will be challenges in terms of convincing stakeholders, and the cost of the technology will be very high. H 4.0 provides numerous opportunities to use new business models such as the platform-based model. It will provide benefits in terms of (1) use of underutilized assets, (2) giving non-strategic assets to ecosystems, (3) modularizing different facets of healthcare services, (4) improving the patient experience, and (5) improving the customer experience, thereby bringing in more customers to the platform, which further improves the value [50,53,56]. Thus, for the successful implementation of H 4.0, improved business models may be unearthed that introduce new ways of making money using digital technologies.

## 5. Discussion

This study identifies ten factors that influence the successful implementation of H 4.0, and examines their impact on the implementation process in two ways. First, it considers the individual impact of each factor on the successful implementation of H 4.0. Second, it explores the interdependence between these factors and how they can collectively contribute to the successful implementation of H 4.0. This section is divided into two subsections to address both aspects.

### 5.1. Individual Impact of CSFs on Successful Implementaiton of H 4.0

This study delineates ten CSFs, which are essential for the successful implementation of H 4.0. The first CSF is digital integration and interconnectedness of the healthcare ecosystem; the key factor here will be to convince the various stakeholders in the healthcare system, including patients, healthcare providers, insurers or payers, and regulatory bodies, as regards the use of H 4.0 technologies to create a more efficient, effective, and patient-centered healthcare system [47,71]. The second CSF is the human-centric automation of healthcare; the key factor here will be the decision as regards which aspects of the healthcare system should have H 4.0 technologies applied while still maintaining the critical role of human decision-making and judgement. The third CSF is that healthcare 4.0 must improve and prioritize the patient experience and ensure that patients are at the center of all healthcare decisions [52,53,54]. The key factor here is that healthcare systems must adopt patient-centric H 4.0 technologies that enable patients to be active participants in their healthcare. The fourth CSF is the use of big data and analytics; the key factor for its success will be how healthcare service providers analyze the large volumes of healthcare data to identify patterns and trends that can be used to improve patient outcomes and experiences and optimize healthcare delivery. The fifth CSF is the use of digital healthcare supply chains; the key factor for success here will be leveraging retail supply chain technologies to improve the efficiency of healthcare delivery, reduce waste, and improve the availability of healthcare products and services. The sixth CSF unearthed in the study is the importance of strategy in the implementation of H 4.0; the key factor for its success will be a well-defined strategy that can help healthcare organizations to maximize the benefits of H 4.0 technologies while minimizing the risks and challenges associated with adoption. By aligning technology investments with organizational goals and objectives, improving collaboration and communication, and ensuring that data are collected and analyzed in a meaningful and actionable way, healthcare organizations can position themselves for the successful implementation of H 4.0. The seventh CSF unearthed in the study is to have a culture to implement H 4.0; the key factor here will be to have a supportive organizational culture that can facilitate the adoption of new technologies, promote collaboration and communication between different stakeholders, and ensure that H 4.0 initiatives are ethical and comply with data privacy. The eighth CSF is to have leaders who can foster a culture of innovation and continuous improvement, address resistance to change, align initiatives with organizational goals, and manage complex and rapidly changing environments, which are essential for the success of H 4.0. The ninth CSF is the importance of employee skills, which are essential for the successful implementation of healthcare 4.0. Healthcare organizations need to invest in employee training and development to ensure that their employees possess the necessary technical and soft skills to effectively utilize and maintain these H 4.0 technologies [20,70]. The last CSF is the importance of the adoption of new business models. Healthcare organizations need to develop innovative and adaptable business models that leverage new technologies and data to improve patient outcomes, reduce costs, and navigate the complex regulatory and legal requirements of healthcare 4.0. By developing new models for financing and reimbursement, promoting collaboration and partnerships, and adapting to changing patient needs and regulatory requirements, healthcare organizations can ensure that healthcare 4.0 is implemented effectively. For successful implementation, all ten CSFs will have interdependence, with impacts on one another. For instance, having a good leader who supports H 4.0 will help in promoting a culture for H 4.0 and devising a strategy for the implementation of H 4.0.

### 5.2. Interdependnace of CSFs for Sucessful Implemenation of H 4.0

Digital integration and interconnections are critical in today’s healthcare industry [47,71]. The ten CSFs are interdependent, requiring consideration of their effects on each other for the successful implementation of H 4.0. With the successful implementation of H 4.0, healthcare providers can connect with patients and other providers more easily and efficiently, resulting in the generation of big data [15,18]. Big data analytics can be used to provide coordinated and integrated care to patients, resulting in an improved patient experience. The use of an appropriate strategy, along with human-centered automation, improves the efficiency of care delivery and creates a patient-centric experience. The patient-centric experience is another factor that is dependent on digital integration and interconnection and human-centered automation. When patients have access to personalized digital tools and technologies, they can participate more actively in their own care. This participation leads to better health outcomes as patients become more engaged and motivated to take care of their health. Moreover, big data and analytics can further help healthcare providers better understand patient needs and preferences, which allows them to peronalize care to meet individual patient needs. Digital healthcare supply chains are also essential in ensuring that patients have access to the healthcare, medications and other supplies they need. When the supply chain is optimized using H 4.0 technologies, patients can receive their medications and supplies quickly and efficiently, which leads to better health outcomes and an improved patient experience. All of these factors are interconnected and interdependent. A well-defined strategy is necessary to ensure that all of these elements are working together to achieve the overall goal of improving patient outcomes. An innovative and adaptable business model is also crucial as it allows healthcare providers to adjust to changes in the industry and meet the evolving needs of patients. It further helps in sustaining the performance of H 4.0. Leadership that fosters innovation and continuous improvement is essential in driving change and ensuring that all the stakeholders are using H 4.0 technologies constantly to improve the patient-centricity and experience. Finally, employee skills development is critical in ensuring that healthcare providers have the knowledge and tools they need to deliver high-quality care using H 4.0 technologies. When employees are well-trained and equipped with the latest technologies and tools, they are better able to meet the needs of patients and provide them with a patient-centric experience. Last but not least, implementing digital integration, driving forward human-centered automation, devising strategies for H 4.0 promotion, and using big data analytics require an organizational culture where the shared values and beliefs of all healthcare stakeholders are centered on improving patient-centricity and the patient experience using digital technologies. Thus, the successful implementation of H 4.0 rests on the important foundation of promoting a culture of H 4.0.

## 6. Scope for Future Research

This study finds ten critical success factors for the successful implementation of H 4.0. None of the studies in the literature explicitly addressed all the factors for the successful implementation of H 4.0, indicating the need for studies that explicitly outline the factors that could impact the implementation of H 4.0.Another interesting area of future research could be qualitatively exploring the success factors in various healthcare subsectors. Studies may examine whether these factors may differ, for example, in biotechnology, pharmaceuticals, equipment, facilities, distribution, and managed healthcare.The healthcare system and ICT technologies vary across countries [89,90]; hence, there is a need for studies to explore whether these critical success factors vary between developing and developed countries.Studies should also quantitively rank the critical success factors so that the importance of these factors can be examined in various settings such as between developed and developing countries. In addition, understanding the relevance of these factors will help in developing a framework for the successful implementation of H 4.0.Future studies should also be targeted on how to develop digital integration and the interconnectedness of the healthcare ecosystem. Studies have examined these in parts [19,47]; however, comprehensive insights as regards developing a framework for integration and interconnectedness are lacking.Human-centric automation design is gaining importance in healthcare systems. Healthcare systems are sociotechnical systems, and hence, to be successful, there is a need for a high level of compatibility between social and technical systems. Future studies should explore the relationship between human-centric automation and the success of H 4.0 implementation.H 4.0 success depends on its ability to improve the patient experience and patient-centricity [53,76]. Future studies should examine how H 4.0 technologies impact the various dimensions of the patient experience. There is also a need for longitudinal studies to understand the time-oriented changes patients experience with the use of digital technologies.

## 7. Conclusions

H 4.0 is an emerging area of research. Although there has been interest in academia in H 4.0, none of the previous studies examined the critical success factors for implementing H 4.0. This study through a systematic review explicates ten critical success factors for the successful implementation of H 4.0. Our findings contribute to the theory of healthcare operations management. First, the study thematically analyzes the literature to explore ten critical success factors for H 4.0 implementation. Second, this study, through bibliometric analyses, describes the development of the body of knowledge on H 4.0 implementation. Third, future research directions are proposed that will support the development of the field of healthcare operations management.

The study finds that the successful implementation of H 4.0 requires the adoption of a patient-centric approach and the alignment of technology investments with organizational goals and objectives. Furthermore, the study highlights the importance of a supportive organizational culture, effective leadership, and employee skills in the successful implementation of H 4.0. The study also emphasizes the need for healthcare organizations to develop innovative and adaptable business models that leverage new technologies and data to improve patient outcomes, reduce costs, and navigate the complex regulatory and legal requirements of healthcare 4.0.

In terms of implications for practice, this study will help healthcare service providers, policymakers and other stakeholders to be sensitive to the critical success factors for H 4.0 implementation. Furthermore, this study will also help healthcare service providers to assess the current state of H 4.0 implementation and how actions can be taken in each of the critical dimensions to successfully implement H 4.0. The study’s scope was limited by the databases it utilized, and its analysis was limited to English-language articles. Additionally, only peer-reviewed research papers were included, which may have excluded other relevant sources of information. As a result, future research should expand its scope to include a wider range of documents.

## Figures and Tables

**Figure 1 ijerph-20-04669-f001:**
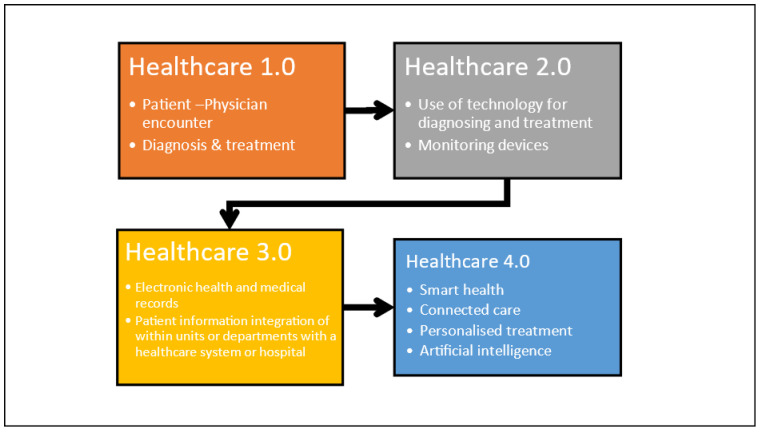
Journey of healthcare from 1.0 to 4.0.

**Figure 2 ijerph-20-04669-f002:**
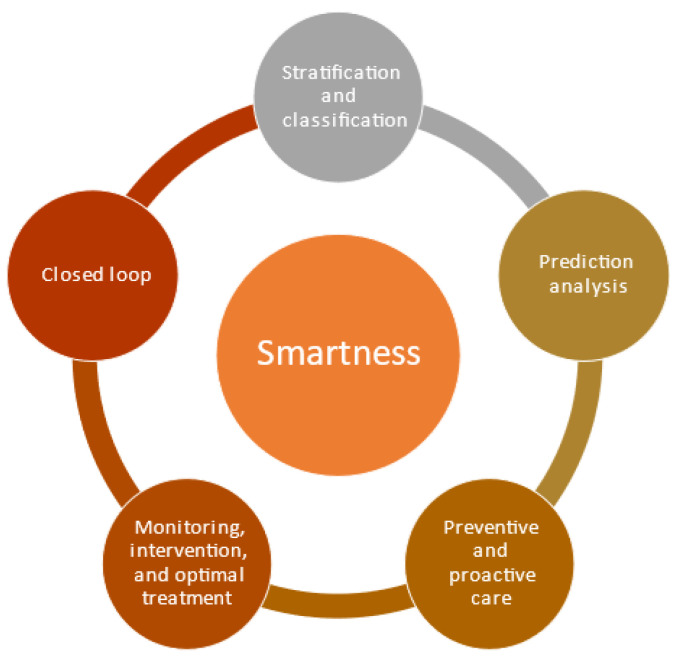
Smartness of H 4.0.

**Figure 3 ijerph-20-04669-f003:**
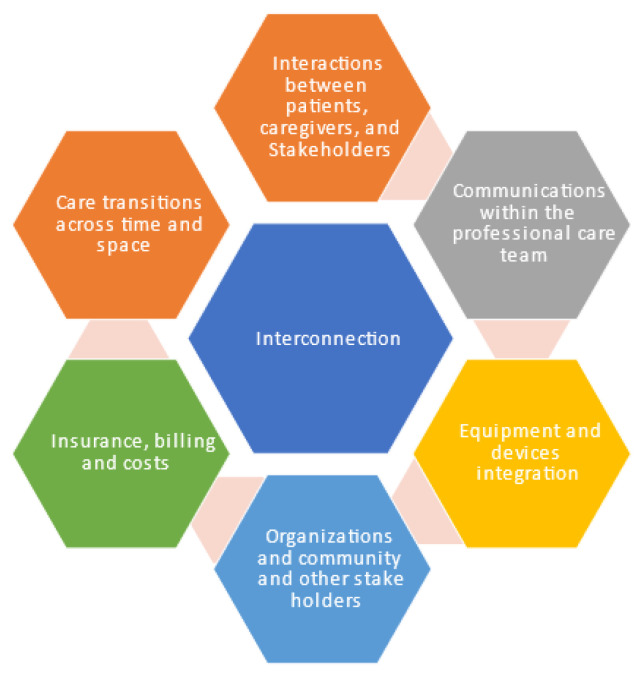
Interconnectedness of H 4.0.

**Figure 4 ijerph-20-04669-f004:**
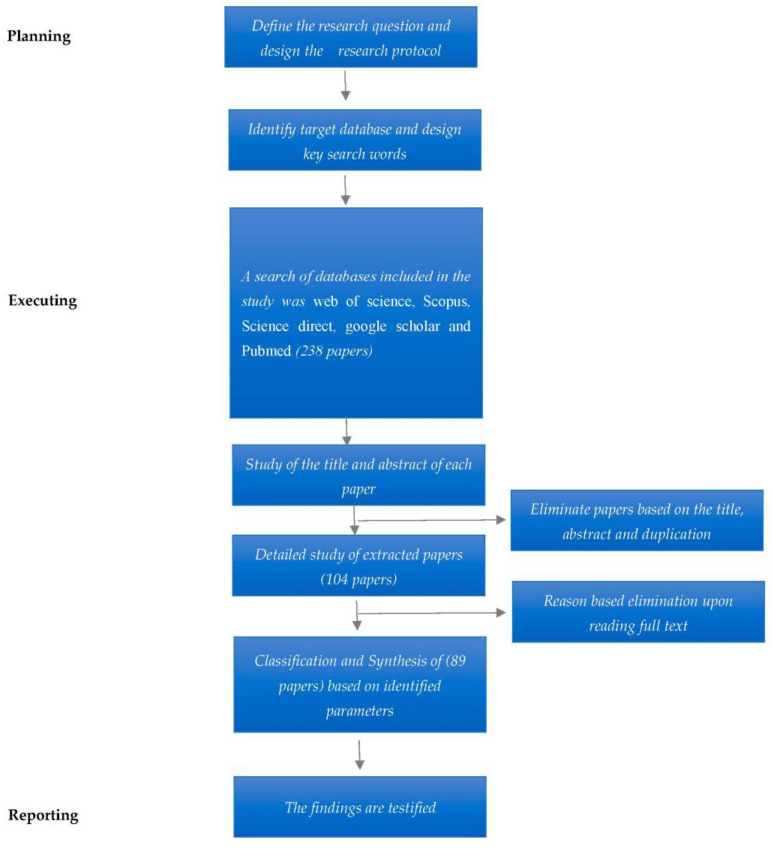
Systematic review methodology.

**Figure 5 ijerph-20-04669-f005:**
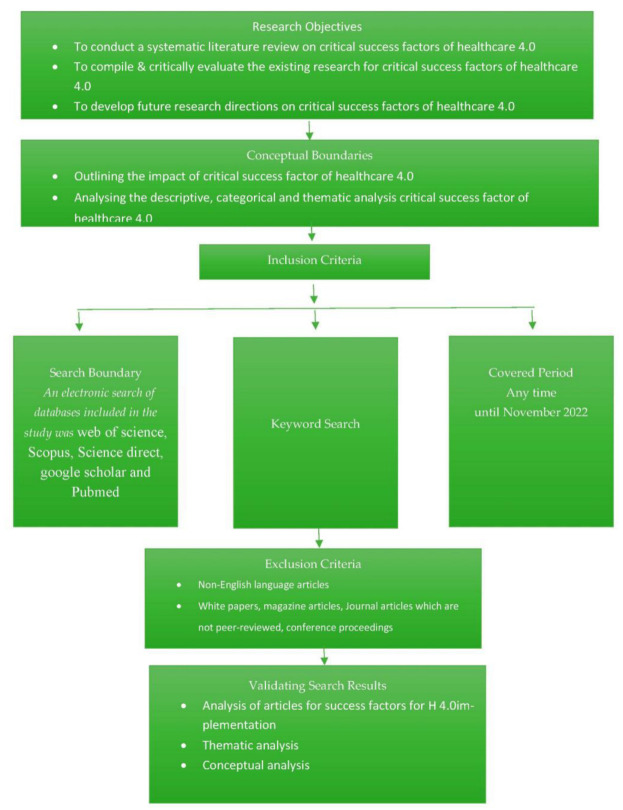
Literature review protocol.

**Figure 6 ijerph-20-04669-f006:**
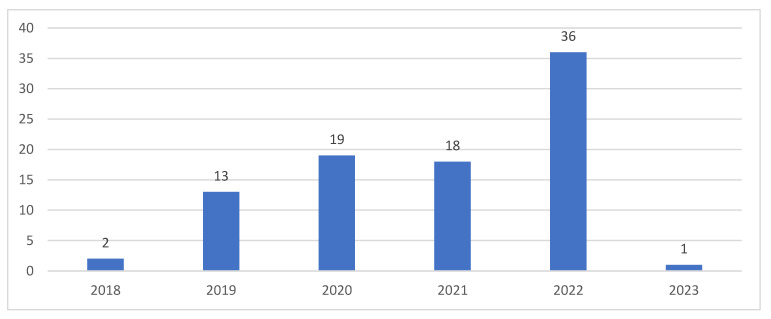
Time trend analysis.

**Figure 7 ijerph-20-04669-f007:**
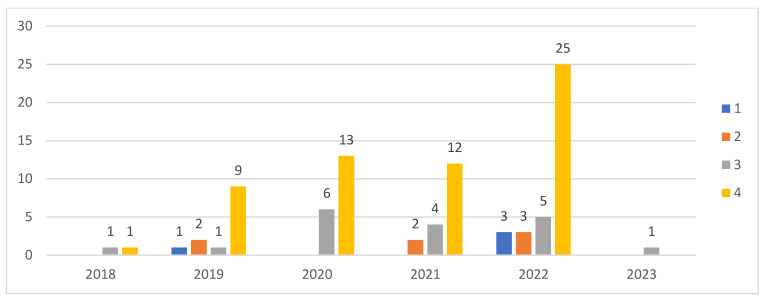
Number of authors.

**Figure 8 ijerph-20-04669-f008:**
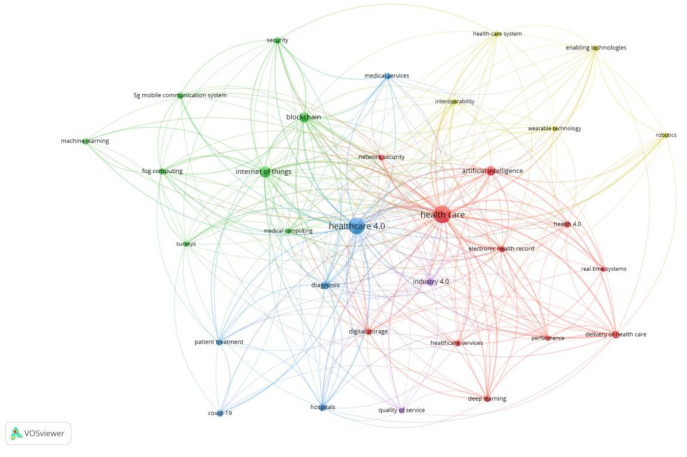
Keyword analysis.

**Figure 9 ijerph-20-04669-f009:**
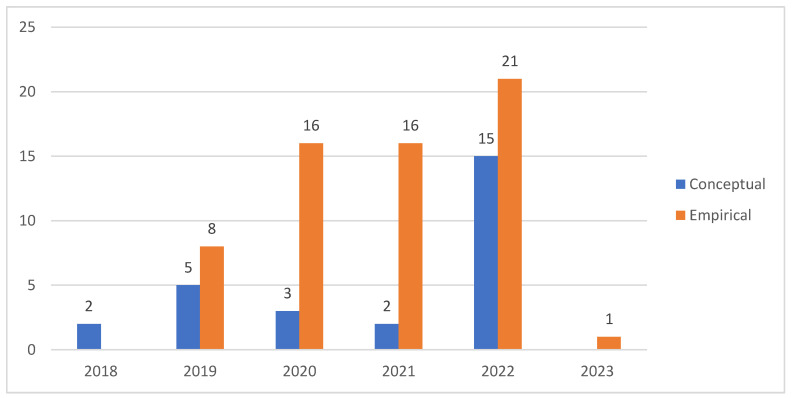
Type of study.

**Table 1 ijerph-20-04669-t001:** Productive authors.

Sr No	Author	Articles	Total Citations
1	Fogliatto F.S.	12	157
2	Kumar N.	11	1038
3	Tanwar S.	11	1361
4	Tortorella G.L.	11	156
5	Tyagi S.	7	979
6	Pang Z.	5	127
7	Vassolo R.	5	100
8	Yang G.	5	127
9	Narayanamurthy G.	4	51
10	Saurin T.A.	4	53
11	Tonetto L.M.	4	53

**Table 2 ijerph-20-04669-t002:** Influential authors.

ID	Author	Documents	Total Citations	CPP
1	Parekh K.	1	339	339
2	Mistry I.	1	273	273
3	Evans R.	2	363	181.5
4	Tyagi S.	7	979	139.86
5	Kumari A.	2	271	135.5
6	Tanwar S.	11	1361	123.73
7	Kumar N.	11	1038	94.36
8	Liu M.	1	90	90
9	Memmi G.	1	90	90
10	Qiu H.	1	90	90
CPP = citations per publication				

**Table 3 ijerph-20-04669-t003:** Top ten journals.

ID	Source	Documents
1	*IEEE J. Biomed. Health Inform.*	5
2	*IEEE Trans. Ind. Inform.*	5
3	*IEEE Access*	4
4	*Comput. Electr. Eng.*	3
5	*IEEE Internet Things J.*	3
6	*Int. J. Prod. Res.*	3
7	*Technol. Forecast. Soc. Change*	3
8	*IEEE Netw.*	2
9	*IEEE Trans. Eng. Manag.*	2
10	*IEEE Trans. Netw. Sci. Eng.*	2

**Table 4 ijerph-20-04669-t004:** Top ten countries.

ID	Country	Documents	Citations
1	India	34	1670
2	Brazil	19	246
3	Australia	16	124
4	China	12	202
5	United Kingdom	12	536
6	United States	11	315
7	Argentina	10	117
8	Sweden	7	162
9	Canada	5	139
10	Chile	5	100

## Data Availability

No new data were created or analyzed in this study. Data sharing is not applicable to this article.

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
