# Peer review of "Critical Success Factors for Successful Implementation of Healthcare 4.0: A Literature Review and Future Research Agenda"

_ijerph, 2023, doi:10.3390/ijerph20054669_

Round 1
Reviewer 1 Report
This paper targets the promising factors for a successful implementation of Healthcare 4.0. However, the article is not appropriate in terms of structure, language and contents. The whole paper is structured as a list of concepts, without a real discussion. The target and contribution to new knowledge in this field remains unclear. Figures and tables are badly presented and not properly described in the text, moreover some are unnecessary. Many unclear and banal sentences, see specific comments.

Reviewer 2 Report
l There are many unexplained italics in the manuscript, which is suggested to be corrected by the authors according to the template. And because the format of the references was different from the journal's regulations, it needs to be revised.
l The introduction is difficult to read in one paragraph, and it is recommended that authors please make appropriate paragraphs.
l The search keywords used by the authors should be described in the Research Method.
l The number of authors published in each article, the number of publications by the author, and the number of citations are not as valuable as analyzing the professional background of the author. It is recommended that the author analyze the professional background of the authors in Table 2 and Table 3.
l Although this manuscript was a systematic review, it should still be written in a paper format. It is recommended that authors make a clear distinction between results and discussions.
Reviewer 3 Report
I appreciate the authors’ efforts into the research of Critical Success Factors for Successful Implementation of Healthcare 4.0: A Literature review and Future research Agenda.
To enhance the quality of the study, the authors must do some revision of their research and pay attention to several important issues:
- The paper should match the format of the journal. Also, the text must be formatted in the same style throughout the article. Currently, part of the text is written normally, and another part is in Italic.
- Section “2.0 Transition from Healthcare 1.0 to 4.0” should be renamed “Literature review”.
- Extensive editing of English language and style is required and authors must pay attention to the punctuation, as there are many double spaces or spaces missing throughout the paper.
- Authors should include some recent works that use bibliometric review, such as:
Mapping Knowledge Area Analysis in E-Learning Systems Based on Cloud Computing. Electronics 2023, 12, 62. https://doi.org/10.3390/electronics12010062.
Bibliometric Analysis of the Green Deal Policies in the Food Chain. Amfiteatru Econ. 2022, 24, 410–428. DOI:10.24818/EA/2022/60/410.
- The tables in the article need to be formatted according to the requirements of the journal.
- The reference list is not formatted according to the MDPI guidelines. Authors should correct this aspect.
- I recommend a different section to be included in the paper: “Findings”, where authors can present and detail the findings and importance of the study results.
- Authors need to explain the limitations of this research in the revised draft.
Round 2
Reviewer 1 Report
The discussion section was included, but still the successful factors are just listed without being compared among each other, their interdependencies are not examined at all. This section would be important in order to highlight the real added value and contribution of the research in the field of healthcare.
Author Response
The discussion section was included, but still the successful factors are just listed without being compared among each other, their interdependencies are not examined at all. This section would be important to highlight the real added value and contribution of the research in the field of healthcare.
Authors reply: We have included a separate section under discussion to highlight the interdependencies of the CSF’s.
Reviewer 2 Report
The discussion section only listed 10 CSFs for healthcare 4.0, but did not address how the different factors really interact and contribute to the implementation of healthcare 4.0. It is suggested that the authors simplify the description of CSF and add more discussion on the implementation and contribution to healthcare 4.0 in the discussion section.
Author Response
The discussion section only listed 10 CSFs for healthcare 4.0 but did not address how the different factors really interact and contribute to the implementation of healthcare 4.0. It is suggested that the authors simplify the description of CSF and add more discussion on the implementation and contribution to healthcare 4.0 in the discussion section.
Authors reply: We have included a separate section under discussion to highlight the interdependencies of the CSF’s.
Reviewer 3 Report
The authors improved the quality and presentation of their research. I recommend the publication after the authors modify the content according to the template of the journal.
Author Response
The authors improved the quality and presentation of their research. I recommend the publication after the authors modify the content according to the template of the journal.
Authors reply we thank the reviewer for the suggestion for recommendation for publication. We have accordingly revised the paper as per the template.